# Current Antimicrobial Susceptibility Trends and Clinical Outcomes of Typhoidal *Salmonella* in a Large Health Authority in British Columbia, Canada

**DOI:** 10.3390/tropicalmed10040108

**Published:** 2025-04-15

**Authors:** Calvin Ka-Fung Lo, Merisa Mok, Cole Schonhofer, Kevin Afra, Shazia Masud

**Affiliations:** 1Department of Pathology and Laboratory Medicine, University of British Columbia, Vancouver, BC V5Z 1M9, Canada; calvinlo66@alumni.ubc.ca (C.K.-F.L.);; 2Antimicrobial Stewardship Program, Fraser Health, Burnaby, BC V5G 2X6, Canada; 3Division of Infectious Diseases, Department of Medicine, University of British Columbia, Vancouver, BC V5Z 3J5, Canada; 4Division of Infectious Diseases, Department of Medicine, Fraser Health, Surrey, BC V3V 1Z2, Canada; 5Antimicrobial Stewardship and Infection Prevention and Control Program, Fraser Health, Surrey, BC V3T 0H1, Canada; 6Department of Pathology and Laboratory Medicine, Surrey Memorial Hospital, Fraser Health, Surrey, BC V3V 1Z2, Canada

**Keywords:** typhoid, paratyphoid, *Salmonella* Typhi, typhoidal *Salmonella*, antimicrobial resistance

## Abstract

Background: From 2018 to 2021, travel-related extensively drug-resistant (XDR) *Salmonella* Typhi was identified in Ontario, Canada. Opportunities remain to characterize typhoidal *Salmonella* antimicrobial susceptibility trends (including multi-drug resistance phenotypes; MDR) within a large health authority in British Columbia, Canada. Methods: This retrospective study included patients with *Salmonella* Typhi or Paratyphi A, B or C bacteremia identified at Fraser Health regional microbiology laboratory from 2018 to 2024. The primary outcome was the proportion of cases with MDR and XDR typhoidal *Salmonella*. Secondary outcomes included annual antimicrobial susceptibility for ampicillin, ceftriaxone, ciprofloxacin, trimethoprim-sulfamethoxazole, ertapenem, meropenem and azithromycin. Clinical outcomes included hospitalization length, and 30-day mortality, clinical cure and infection relapse. Results: Among 271 patients, most were previously healthy and recently travelled. There were extended spectrum beta-lactamase (1.1%) and MDR (1.5%) typhoidal *Salmonella*, with no XDR cases observed. In 2024, ciprofloxacin resistance was 96% while susceptibility rates were high for other studied antimicrobials. Within 30 days, no deaths were reported; however, six patients (3%) had infection relapse. Conclusions: Currently, in British Columbia, MDR typhoidal *Salmonella* remains rare. Empiric ciprofloxacin should be avoided due to persistently high resistance rates. With ongoing travel patterns, it is beneficial for institutions to continue typhoidal *Salmonella* antimicrobial susceptibility surveillance, and travelers should seek pre-travel health assessments.

## 1. Introduction

Enteric fever, including typhoid and paratyphoid fever, is a systemic febrile illness caused by typhoidal *Salmonella*. Pathogens include *Salmonella enterica* subspecies *enterica* serovars Typhi and Paratyphi A, B and C, and their disease impact remains a significant global health concern [1]. In 2021, an estimated 9.3 million global cases resulted in 107,500 deaths, with the highest disease burden in children in South Asia and Southeast Asia [1]. Although of lower incidence, cases of enteric fever have been reported from developed countries. For instance, Canada averaged 160 annual reported cases between 2018 and 2022 (0.42 cases/100,000 population), with 239 reported cases (0.61/100,000 population) in 2022 alone [2]. Most Canadian cases were linked to travel to endemic countries; however, a local typhoid fever outbreak in Ottawa, Ontario was linked to an asymptomatic chronic *S.* Typhi carrier who worked in food handling [3].

*S.* Typhi and *S.* Paratyphi A, B, and C differ from other non-typhoidal *Salmonella* serovars as humans are their sole reservoir [4]. Person-to-person transmission occurs fecal-orally with endemic circulation in communities without sufficient infrastructure for clean water sanitation [4]. After ingestion, its incubation period typically ranges from 1 to 4 weeks [5,6]. Classically, the cases of enteric fever present with progressive fever, malaise, headache, abdominal pain with either constipation or diarrhea, and a characteristic pink macular rash (rose spots) [6]. Severe complications of untreated disease include gastrointestinal hemorrhage and perforation, encephalopathy and septic shock. The mortality rate falls below 1% if the disease is identified and treated early with effective antibiotics, but may be 10 to 30% in untreated cases [6]. Additionally, 2 to 5% of survivors develop a chronic asymptomatic carrier state that remains capable of transmitting *S.* Typhi [7]. Enteric fever is generally diagnosed when either *S.* Typhi or *S.* Paratyphi is detected in blood or stool cultures in patients with compatible febrile illness and exposure history.

The emergence and spread of antimicrobial resistance among typhoidal *Salmonella* spp. is concerning and has complicated the antimicrobial treatment of enteric fever [5]. Multi-drug-resistant (MDR) strains, which are resistant to ampicillin, trimethoprim-sulfamethoxazole and chloramphenicol, first emerged in the 1980s. As such, treatment options are more limited and may involve fluoroquinolones or third-generation cephalosporins (e.g., ceftriaxone). Increased empiric fluoroquinolone usage globally propagated further antimicrobial resistance [5]. Extensively drug-resistant (XDR) *S.* Typhi include MDR resistance patterns but are also resistant to fluoroquinolones and third-generation cephalosporins, further limiting empiric options and increasing carbapenem (e.g., meropenem) and azithromycin usage. XDR *S.* Typhi led to a large outbreak in Pakistan since 2016 and has subsequently been identified globally [5,8,9,10,11,12,13]. In Canada, the first XDR *S.* Typhi case was reported in Toronto, Ontario in 2018 after a pediatric patient travelled to Pakistan [13]. Since then, XDR cases linked to Pakistan travel have been reported in Ontario, Canada from 2018 to 2021 [10,11].

With recent travel and immigration patterns in British Columbia, Canada before and after the global COVID-19 pandemic, there remains an opportunity to review current antimicrobial resistance trends for typhoid and paratyphoid cases. In this retrospective study, we describe the antimicrobial susceptibility patterns of *S.* Typhi and *S.* Paratyphi blood stream isolates identified at a regional microbiology laboratory in British Columbia, Canada, from 2018 to 2024. This study took place in the Fraser Health Authority region, which is one of the fastest growing Canadian communities and serves one in three British Columbians [14]. Relevant 30-day clinical outcomes are also described, and further details are in the Materials and Methods section (Section 2).

## 2. Materials and Methods

Study Design, Setting, and Approval: This retrospective review was conducted at Fraser Health Authority, which includes 13 acute care hospitals in British Columbia, Canada (Appendix A, Table A1). The study received ethics and institutional research approval from the University of British Columbia Clinical Research Ethics Board and Fraser Health Research Ethics Board.

Inclusion and Exclusion Criteria: The regional microbiology laboratory data were used to identify patients who presented to one of 13 acute care Fraser Health hospitals and had positive blood cultures with either *Salmonella* Typhi or Paratyphi A, B or C from 1 January 2018 to 31 December 2024. Duplicate isolates from the same patient were excluded, and only the first isolate was included for analysis. Antimicrobial susceptibility testing was performed using the Vitek 2 (bioMérieux) automated susceptibility system, with the exception for azithromycin susceptibility being tested by Kirby-Bauer disk diffusion method. The study period was chosen to observe trends after XDR *S.* Typhi was first identified in Canada in 2018 [13] and to capture antimicrobial susceptibility trends of typhoidal *Salmonella* pre- and post-COVID-19 pandemic.

Data Collection and Analysis: Data variables were extracted from laboratory information system and electronic medical records. Variables included patient age, gender, Charlson comorbidity index score, recent travel history, primary acute care service, infectious diseases consultation, concurrent infection(s), symptoms, year of infection, pathogen isolated, antimicrobial susceptibility profile, treatment intervention and the clinical outcomes detailed below. Standard descriptive analysis (including median, inter-quartile range and percentage proportions) was performed using Excel (Redmond, Washington).

Outcomes: The primary outcome was the proportion of MDR and XDR typhoidal *Salmonella* cases from 2018 to 2024. For secondary outcomes, we included annual distributions of categorical susceptibility to each agent: ampicillin, ceftriaxone, ciprofloxacin, trimethoprim-sulfamethoxazole (TMP-SMX), meropenem, ertapenem and azithromycin (if available) from 2018 to 2024. Susceptibility interpretations were based on Clinical and Laboratory Standards Institute (CLSI) M100 breakpoints [15]. Additionally, hospitalization length (in days), along with mortality, clinical cure and infection relapse within 30 days, were included for clinical outcomes.

## 3. Results

### 3.1. Enteric Fever Incidence and Patients’ Baseline Characteristics

Between 2018 and 2024, there were 583 typhoidal *Salmonella* blood culture isolates. After excluding duplicate isolates, there were 271 initial isolates from patients (Table 1). A total of 166 blood cultures (61%) were positive for *S.* Typhi, while there were 98 cases (36%) of *S.* Paratyphi A and seven cases (3%) of *S.* Paratyphi B. There were no *S.* Paratyphi C cases identified. About one-third of enteric fever cases occurred from 2018 to 2019 (Appendix A, Table A1). When the COVID-19 pandemic occurred, the total number of cases significantly decreased to 24 cases between 2020 and 2021. From 2022 to 2024, annual incidence increased and exceeded pre-pandemic rates (Appendix A, Table A1).

Detailed patient baseline characteristics are described in Table 1 and Appendix A, Table A1. Overall, patients had a median age of 29 years (IQR 20–42) and were previously healthy (median Charlson comorbidity index score of 0). Nearly all patients (96%) had recently travelled within the last 3 months, for reasons including leisure, business, emergency travel or immigration. Most travelers visited India (90%), and the next most frequent country travelled was Pakistan (3.5%). Other travel regions included Southeast Asia, Latin and North America (including Mexico and Southeastern United States), the Middle East and East Asia. About a quarter of patients had documented possible exposure or consumption of contaminated food or water during their travel; sources were thought to be from insufficient sanitary water access, undercooked meats and/or street food. Fever was the most common symptom (97.8%) followed by gastrointestinal symptoms (73.5%). Most cases were mainly managed medically either with short hospital admissions or through the emergency department and outpatient antibiotic therapy program (98.1%). Four patients (1.5%) had sepsis or septic shock. Two patients required prompt surgical intervention for severe gastrointestinal perforation (0.74%). One of these patients was previously healthy, had a prolonged hospital stay with septic shock, multiple surgical procedures and prolonged antibiotic treatment, but unfortunately passed away in hospital in 3 months from a non-infectious factor. The other patient was immunocompromised and had surgery and prolonged hospitalization. There was also one severe case of *S.* Paratyphi B meningitis and bloodstream infection in a 2-month-old patient with no known travel history or sick contacts. Infectious Diseases services were consulted in 202 cases (75%) for diagnostic and treatment advice. The median total duration of effective antibiotics was 12 days (IQR 10 to 14).

### 3.2. Antimicrobial Susceptibility Trends

#### 3.2.1. Key Findings

During the 7-year period of typhoidal *Salmonella* cases, ESBL was identified in 1.1% of cases and MDR was identified in 1.5% of cases (Table 1). These resistance phenotypes were only identified in *S.* Typhi. There were no XDR typhoidal *Salmonella* cases identified. Ciprofloxacin resistance, which was based on isolates with reported resistance (i.e., minimum inhibitory concentration (MIC) ≥ 1 µg/mL) or intermediate susceptibility (i.e., MIC of 0.5 µg/mL), remained high (i.e., 92% in 2018 and 96% in 2024) (Table 2 and Table 3).

#### 3.2.2. *S.* Typhi Antimicrobial Susceptibility

In 166 *S.* Typhi cases from 2018 to 2024, ciprofloxacin had the highest annual resistance rate (Table 2). By 2020, the susceptibility rate had decreased to 0%. While the susceptibility rate increased to 16.7% in 2021 (total of six isolates), it again decreased to 0% by 2024. In contrast, most isolates were susceptible to ampicillin by 2024. Ceftriaxone had high susceptibility (nearly 100% in all years), except in three ESBL phenotype isolates that were identified during the study period (1.8% of *S.* Typhi cases) (Table 1 and Table 2). TMP-SMX was generally susceptible (mostly above 90%), except when four MDR isolates were identified (in 2.4% of *S.* Typhi cases) (Table 1 and Table 2). After azithromycin became routinely tested in 2019, the data showed high susceptibility rates except for one non-susceptible *S.* Typhi isolate in 2022 (Table 2). This patient, who had azithromycin-resistant and ciprofloxacin-resistant *S.* Typhi, had recently travelled to India, and it was unclear if the patient had ingested contaminated water. All isolates tested susceptible to meropenem and ertapenem across the 7-year study period.

#### 3.2.3. *S.* Paratyphi Antimicrobial Susceptibility

In 105 *S.* Paratyphi A and B bacteremia cases, all had high susceptibility to ceftriaxone, TMP-SMX, ertapenem, meropenem and azithromycin (Table 3). While ampicillin susceptibility had 38% and 60% susceptibility in thirteen isolates (in 2018) and twenty isolates (in 2019), respectively, susceptibility rates increased to 100% after the COVID-19 pandemic. Similar to *S.* Typhi, *S.* Paratyphi resistance to ciprofloxacin was high across the study period (Table 3). There was only one azithromycin-resistant *S.* Paratyphi A case, and this patient was managed in the community by the patient’s family physician.

#### 3.2.4. Impact of Carbapenemase Colonization on Antimicrobial Susceptibility Results

While 7% of patients (19/271) were colonized with NDM and/or OXA-48 carbapenemases (Appendix A, Table A1), carbapenem resistance was not identified in *S.* Typhi and *S.* Paratyphi blood cultures.

### 3.3. Clinical Outcomes

Clinical outcomes, summarized in Table 4, were identified using the health authority electronic medical record. Seventy-nine cases (29%) did not have healthcare encounters one month after initial presentation and, as such, it was not possible to determine the clinical outcomes of our study. Of the available data (192/271 cases), there was no reported mortality within 30 days of diagnosis. However, there were six cases of relapsed infection within 30 days; two of these cases occurred in the setting of ciprofloxacin step-down therapy in typhoid infection with intermediate susceptibility to ciprofloxacin.

Clinical cure within 30 days was 97% (186/192 cases). Among the six cases with delayed clinical cure, two patients with meningitis and gastrointestinal perforation, respectively, were previously discussed. The third patient had ongoing investigations for liver and biliary tract infection, which was later determined to be malignancy related. Three other patients had been initially treated but developed recurrent enteric fever within 30 days. Overall, the median hospitalization length was 1 day (IQR 1 to 4 days) including emergency department assessment. This duration did not include subsequent outpatient follow-up visits (e.g., infectious diseases clinic and/or outpatient intravenous antibiotic program).

## 4. Discussion

Most published typhoid resistance studies in Canada are available from the Ontario province [10,11,13,16]. To our knowledge, this is the first study analyzing typhoid and paratyphoid antimicrobial resistance and clinical outcomes in British Columbia, Canada. Fraser Health is the largest regional health authority in British Columbia and serves one in three British Columbians [14]. In 2023 alone, Fraser Health hospitals had 118 typhoidal *Salmonella* isolates (from 63 patients), which was nearly a third of the number of isolates reported by the Canadian Integrated Program for Antimicrobial Resistance Surveillance (CIPARS) surveillance program (i.e., 441 isolates) [17].

In our 7-year study period, most patients were previously healthy without medical comorbidities, young (median age 29), and had recently travelled to South Asia within the past 3 months before clinically presenting with enteric fever. Exposure to contaminated food or water while travelling in typhoid-endemic areas is a significant risk factor for *Salmonella* infection. Pre-travel vaccination and counselling (e.g., safe food handling) can prevent illness in typhoid-endemic areas (e.g., South Asia) [18,19]. It is important to note that the typhoid vaccine has variable efficacies ranging from 43 to 70% depending on formulation (e.g., oral live attenuated versus intramuscular inactivated); furthermore, it does not prevent against paratyphoid infections, which would have a similar acquisition route as typhoid itself. There is ongoing work in progress for approving typhoid conjugate vaccines in developed countries, which are currently undergoing the WHO pre-qualification process [18,20].

While most study patients did not have severe complications or prolonged hospitalization, progressive fever and ongoing gastroenteritis symptoms can persist in the outpatient setting. Furthermore, the study’s severe cases (e.g., typhoid-related intestinal perforation, septic shock, *S.* Paratyphi meningitis with seizures) demonstrate that severe complications of enteric fever, while rare in developed countries, do exist. Prompt diagnosis and treatment in patients with a history of travel and pathogen exposure are crucial in reducing complications.

In our study’s enteric fever relapse cases, one contributing factor may have been ciprofloxacin usage in the setting of confirmed isolates with intermediate ciprofloxacin susceptibility. In those situations, there is strong debate about whether adequate MIC levels are achieved with existing dosing regimens. With high rates of ciprofloxacin resistance circulating globally, ciprofloxacin should generally be avoided unless the patient has confirmed susceptibility results. The widespread ciprofloxacin resistance observed in our study for both *S.* Typhi and *S.* Paratyphi isolates was not surprising based on travel history (e.g., South Asia) and national CIPARS data [17]. High rates of ciprofloxacin resistance from South Asia have been identified [21]. Although *S.* Typhi isolates in our study showed improved ciprofloxacin susceptibility (i.e., to 16.7%) in 2021, this finding should be cautiously interpreted in the setting of low isolate numbers and restricted travel during the global COVID-19 pandemic; this outlier likely does not reflect our local health authority’s ciprofloxacin susceptibility trend. It is reassuring that most of our antimicrobial susceptibility trends are similar to national CIPARS public data, with the exception of rising ampicillin resistance nationally.

Reassuringly, ESBL and MDR isolates were low in our patient population, despite the majority of cases arising from areas of high antimicrobial resistance. Furthermore, our 7-year study from 2018 to 2024 did not identify XDR-phenotype cases across Fraser Health hospitals. XDR typhoid was first reported in Canada in summer 2018, after the pediatric patient returned from Karachi, Pakistan to Toronto, Ontario [13]. Antimicrobial susceptibility testing confirmed resistance against all first-line agents and susceptibility to meropenem and azithromycin. Subsequently, further XDR *S.* Typhi cases were also identified in Ontario between 2018 and 2021 [10,11]. Although the last XDR isolate in Canada was documented in 2023, no other cases have since been reported in our national CIPARS database [17]. To our knowledge, there has not been a confirmed case of XDR *S.* Typhi or *S.* Paratyphi identified in British Columbia as of March 2025. Although our cases did travel to endemic countries with documented XDR typhoidal *Salmonella*, a possible hypothesis may be that the returning travelers in our study may not have travelled to regions with active circulating outbreaks in South Asia.

Our study had some limitations due to the retrospective study design and available data. Susceptibility results were based on categorical description as MIC values were not readily available across all study period years. However, the antimicrobial breakpoints for *Salmonella* spp. have not changed since 2018, except for formatting changes in the CLSI M100 document (34th edition) in 2024 to separate *Salmonella* and *Shigella* from other Enterobacterales [15]. Hence, categorical susceptibility interpretations did not necessarily change during our research time range (e.g., *S.* Typhi reported as susceptible to ceftriaxone in 2018 would still be considered susceptible in 2024). While the Fraser Health electronic medical record had access to all acute care site information, a key limitation was that other information sources (e.g., records outside of local health authority) were not accessible to review clinical outcomes. Nevertheless, available clinical transcriptions post-infection were carefully reviewed to identify enteric fever recurrence and complications. Lastly, our study patients were identified based on blood cultures only. Bacteremia is typically the most commonly encountered manifestation in enteric fever and, therefore, blood cultures are a reference standard for diagnosis; however, this study may have missed some patients who either were seen in a healthcare institution outside of Fraser Health Authority or already received effective treatment before blood cultures were collected [5,6].

Future research may involve ongoing antimicrobial susceptibility surveillance for highly used enteric fever antibiotics (e.g., azithromycin, ceftriaxone) in British Columbia. An opportunity may be for institutions to collaborate with a global surveillance platform network (e.g., GeoSentinel) to gather real-time data in improving clinical outcomes.

## 5. Conclusions

From 2018 to 2024, the incidence of MDR typhoidal *Salmonella* bacteremia remains very low in a large health authority in British Columbia, Canada. Travel to countries with fluoroquinolone resistance for typhoidal *Salmonella* has led to high ciprofloxacin resistance rates (96%) in our health authority. Reassuringly, no travel-related XDR strains were identified since the first Canadian XDR case occurred in 2018 in Ontario. However, with ongoing travel patterns post-pandemic, it is beneficial for travelers to seek pre-travel health assessments to mitigate acquisition risk. Laboratory institutions should consider conducting ongoing surveillance on antibiotic susceptibility rates for typhoidal *Salmonella* isolates.

## Figures and Tables

**Table 1 tropicalmed-10-00108-t001:** Baseline characteristics of included patients with *Salmonella* Typhi or Paratyphi A, B or C Bacteremia (n = 271).

Characteristics	N (%) of Cases Unless Specified
Median age in years (IQR)	29 (20 to 42)
Age group:	
2 years or less	11 (4%)
3–17 years	48 (18%)
18–29 years	83 (31%)
30–39 years	53 (19%)
40–49 years	32 (12%)
50–59 years	25 (9%)
60 years or above	19 (7%)
Sex, male	137 (51%)
Median Charlson Comorbidity Index Score (IQR)	0 (0 to 0)
Recent travel history within last 3 months	259 (96%)
South Asia	247
India	233
Pakistan	9
Other	5
Southeast Asia	18
Latin America, North America, Middle East or East Asia	16
Potentially consumed contaminated food or water during travel	75 (28%)
Primary acute care service	
Medical	266 (98.1%)
Surgical	1 (0.4%)
Intensive care	4 (1.5%)
Infectious Diseases consultation	202 (75%)
Presentation ^1^	
Fever	262 (97.8%)
Chills	87 (32.5%)
Malaise	65 (24.3%)
Headache	96 (35.8%)
Confusion	6 (2.2%)
Tachycardia	56 (20.9%)
Hypotension	31 (11.6%)
Tachypnea	8 (3.0%)
Abdominal pain, constipation, diarrhea, nausea or vomiting	197 (73.5%)
Rash	8 (3.0%)
Joint pain	8 (3.0%)
Leukocytosis or leukopenia	44 (16.4%)
Thrombocytopenia	10 (3.7%)
Anemia	10 (3.7%)
Elevated liver function tests	63 (23.5%)
Sepsis or septic shock	4 (1.5%)
Meningitis	1 (0.4%)
Intestinal perforation	2 (0.74%)
Blood culture with	
*Salmonella* Typhi	166 (61%)
*Salmonella* Paratyphi A	98 (36%)
*Salmonella* Paratyphi B	7 (3%)
*Salmonella* Paratyphi C	0 (0%)
Typhoid drug resistance phenotype ^2^	
Extended spectrum beta-lactamase (ESBL)	3 (1.1%)
Multi-drug-resistant (MDR)	4 (1.5%)
Extensively drug-resistant (XDR)	0 (0%)
Median total duration of effective antibiotics (IQR) ^3^	12 (10 to 14)
Surgical intervention for intestinal perforation	2 (0.74%)

^1^ Based on available data from 268 patients. ^2^ Drug resistance phenotypes (e.g., ESBL and MDR) were observed in *S.* Typhi isolates only. ^3^ Based on available data from 243 patients.

**Table 2 tropicalmed-10-00108-t002:** Antimicrobial susceptibility for *Salmonella* Typhi blood cultures, 2018–2024 ^1,2^.

Antimicrobial	Susceptibility Rate (%) Stratified by Year						
	2018	2019	2020	2021	2022	2023	2024
**Ampicillin**	**92%**	**100%**	**90%**	**67%**	**97%**	**100%**	**95%**
**Ceftriaxone**	**96%**	**100%**	**100%**	**83%**	**100%**	**100%**	**98%**
**Ciprofloxacin**	**12%**	**8%**	**0%**	**16.7%**	**3%**	**11%**	**0%**
**TMP-SMX**	**96%**	**100%**	**90%**	**83%**	**97%**	**100%**	**98%**
**Ertapenem**	**100%**	**100%**	**100%**	**100%**	**100%**	**100%**	**100%**
**Meropenem**	**100%**	**100%**	**100%**	**100%**	**100%**	**100%**	**100%**
**Azithromycin**	**No data**	**100%**	**100%**	**100%**	**97%**	**100%**	**100%**

^1^ Data based on first isolate of the specific organism from any patient. ^2^
For all listed antibiotics except for azithromycin, susceptibility was tested in the total number of isolates: 25 (in 2018), 24 (in 2019), 10 (in 2020), 6 (in 2021), 33 (in 2022), 27 (in 2023), 41 (in 2024). For azithromycin, susceptibility was tested in the total number of isolates: 11 (in 2019), 10 (in 2020), 5 (in 2021), 32 (in 2022), 27 (in 2023), 40 (in 2024).

**Table 3 tropicalmed-10-00108-t003:** Antimicrobial susceptibility for *Salmonella* Paratyphi A and B blood cultures, 2018–2024 ^1,2^.

Antimicrobial	Susceptibility Rate (%) Stratified by Year						
	2018	2019	2020	2021	2022	2023	2024
**Ampicillin**	**38%**	**60%**	**100%**	**100%**	**100%**	**100%**	**100%**
**Ceftriaxone**	**100%**	**100%**	**100%**	**100%**	**100%**	**100%**	**100%**
**Ciprofloxacin**	**0%**	**10%**	**17%**	**0%**	**7%**	**3%**	**15%**
**TMP-SMX**	**100%**	**100%**	**100%**	**100%**	**100%**	**100%**	**100%**
**Ertapenem**	**100%**	**100%**	**100%**	**100%**	**100%**	**100%**	**100%**
**Meropenem**	**100%**	**100%**	**100%**	**100%**	**100%**	**100%**	**100%**
**Azithromycin**	**No data**	**100%**	**100%**	**100%**	**100%**	**97%**	**100%**

^1^ Data based on first isolate of the specific organism from any patient. ^2^
For all listed antibiotics except for azithromycin, susceptibility was tested in the total number of isolates: 13 (in 2018), 20 (in 2019), 6 (in 2020), 2 (in 2021), 15 (in 2022), 36 (in 2023), 13 (in 2024). For azithromycin, susceptibility was tested in the total number of isolates: 5 (in 2019), 5 (in 2020), 2 (in 2021), 13 (in 2022), 32 (in 2023), 9 (in 2024).

**Table 4 tropicalmed-10-00108-t004:** Clinical outcomes for enteric fever cases in Fraser Health, 2018–2024.

Outcome	n (%) Unless Otherwise Specified
Mortality within 30 days ^1^	0 (0%)
Clinical cure within 30 days ^1^	186 (97%)
Relapse of infection within 30 days ^1^	6 (3%)
Median hospitalization length (days, IQR)	1 (1 to 4 days)

^1^ Based on available data from 192 cases.

## Data Availability

The original contributions presented in this study are included in the article. Further inquiries can be directed to the corresponding author.

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
