# Peer review of "Current Antimicrobial Susceptibility Trends and Clinical Outcomes of Typhoidal Salmonella in a Large Health Authority in British Columbia, Canada"

_tropicalmed, 2025, doi:10.3390/tropicalmed10040108_

Round 1

Reviewer 1 Report

Comments and Suggestions for Authors

Title: Current antimicrobial susceptibility trends and clinical outcomes of typhoidal Salmonella in a large health authority in British Columbia, Canada

This manuscript is well-written by the authors. I do believe that if they can improve the manuscripts following all comments. It might have a chance to publish in the journal.

Comments

  1. Abstract: The authors conclude about MDR typhoidal Salmonella in the conclusion. Please add MDR in the background of the abstract.
  2. Keywords: Actually, 5 keywords are enough.
  3. Introduction is well written.
  4. Line 89-93: Please add an ethical approval no.
  5. Do the authors perform a statistical analysis? Please add some information the method section.
  6. Line 128-130: The authors mentioned that “When the COVID-19 pandemic occurred, the total number of cases significantly decreased to 24 cases between 2020 and 2021.” However, the authors should mention this COVID-19 pandemic in the introduction as well as method.
  7. Table 1: The name of the first column should be “Characteristics”.
  8. Table 1: The name of the second column should be “numbers of samples or numbers of isolates”.
  9. Line 161: “3.2.1. Key Findings”; This section should be combined with “Antimicrobial Susceptibility”.
  10. Please add the information of the discussion. Try to compare the results (the author’s hypothesis) with other finding by other researchers.
  11. Please delete some introduction and result sentences in the discussion part.

Author Response

Comment 1: Abstract: The authors conclude about MDR typhoidal Salmonellain the conclusion. Please add MDR in the background of the abstract.

Response 1: Thank you for reading over our abstract. We have added the term “MDR” earlier in the background of our revised abstract (see new Line 21-22).

Comment 2: Keywords: Actually, 5 keywords are enough.

Response 2: Thank you kindly for your suggestion. We agree that 5 keywords are sufficient, and subsequently removed the following keywords: “AMR”, “MDR”, “fluoroquinolones”, “antibiotic”. We also changed Salmonella Paratyphi to “typhoidal Salmonella”.

Comment 3: Introduction is well written.

Response 3: Thank you very kindly for your compliment. We hope the content is to your liking and will be of great interest for our future readers on this clinically important topic.

Comment 4: Line 89-93: Please add an ethical approval no.

Response 4: Thank you for pointing out this section. We have now inserted our ethics approval number in Line 333.

In this section, we have already clarified that our study received ethics and institutional research approval from both University of British Columbia Clinical Research Ethics Board, along with Fraser Health Research Ethics Board (refer to new Lines 91-93).

Comment 5: Do the authors perform a statistical analysis? Please add some information the method section.

Response 5: Thank you for clarifying this. Yes, we have performed a statistical analysis; however, that was limited to descriptive statistics (including proportion of susceptibility to each antimicrobial, median age, and proportions of specific clinical features/risk factors for acquiring typhoidal Salmonella). We have added further information to new Lines 111-112.

As detailed in our study objectives, our intent was to provide a descriptive overview of the antimicrobial susceptibility trends over a 7-year period for Salmonella Typhi and Salmonella Paratyphi isolates. Therefore, we did not compute more complex statistical analysis (e.g., regression models) based on our study objectives.

Comment 6: Line 128-130: The authors mentioned that “When the COVID-19 pandemic occurred, the total number of cases significantly decreased to 24 cases between 2020 and 2021.” However, the authors should mention this COVID-19 pandemic in the introduction as well as method.

Response 6: Thank you for your feedback. To introduce the COVID-19 pandemic component earlier in our manuscript, we now mentioned it both in Lines 79-80 and Lines 103-104.

Comment 7: Table 1: The name of the first column should be “Characteristics”.

Response 7: Thank you for your feedback, we have addressed and corrected the name as suggested.

Comment 8: Table 1: The name of the second column should be “numbers of samples or numbers of isolates”.

Response 8: Thank you for pointing this out. We corrected the second column as indicated with a slight modification to “number (%) of cases”. As indicated in our Inclusion Criteria, Line 98-99, only the first isolate of each patient case was included for analysis to avoid duplicate counting.

Comment 9: Line 161: “3.2.1. Key Findings”; This section should be combined with “Antimicrobial Susceptibility”.

Response 9: Thank you kindly for your suggestion. Given the subsections 3.2.2 and 3.2.3 have more specific details for Salmonella Typhi and Salmonella Paratyphi antimicrobial susceptibilities separately, our group consensus is that it is best to keep the Key Findings (i.e., Section 3.2.1) as a separate subsection. The Key Findings provided a concise high-level overview of detected resistant phenotypes of interest (e.g., ESBL, MDR, XDR) and overall proportion of fluoroquinolone resistance.

Comment 10: Please add the information of the discussion. Try to compare the results (the author’s hypothesis) with other finding by other researchers.

Response 10: Thank you for reviewing our Discussion section. We have carefully reviewed our Discussion. In our opinion, we have already compared our results to existing findings from other researchers and national surveillance database (CIPARS) for Canada (Lines 280-290). We had also provided an overview of the emergence of XDR typhoid in Canada from travel-related cases. (Lines 280-290). Interestingly, our local study based in British Columbia did not identify any XDR cases of typhoidal Salmonella.

Comment 11: Please delete some introduction and result sentences in the discussion part.

Response 11: Thank you for your suggestion. We have carefully reviewed our Discussion section and would say that we have already condensed down the Introduction and Results summary to a limited section of 10 lines (Lines 231-241). We carefully considered how readers may need this necessary information to better understand the flow and content of the Discussion section. Please kindly let us know if there is any specific sentence that you will like us to further modify.

Reviewer 2 Report

Comments and Suggestions for Authors

line 287: Please clarify limitation, while the study mentions the retrospective design and the absence of MIC values for some years, it could also discuss any potential biases or gaps in the data, especially regarding the generalizability of findings to other regions in Canada or beyond.

I recomand further Research on XDR S. Typhi, given that the study didn’t observe XDR cases, it would be useful to explore in the discussion section the potential reasons for the absence of such cases in British Columbia compared to Ontario and other regions.

Please expand on Pre-travel Health Assessments. The suggestions for pre-travel health assessments is important; however, providing more specific recommendations or data on how effective such programs have been elsewhere could strengthen this recommendation.

I would like to recommend that you cite the following paper in your study, as it addresses crucial aspects of nosocomial infections and their impact, which aligns with the theme of your research:
This paper provides valuable information on the distribution and frequency of nosocomial infections in healthcare settings. Including it in your references would enhance the comprehensiveness of your literature review and strengthen the foundation for the objectives and conclusions of your study.  

Stanga, L.C.; Vaduba, D.M.B.; Grigoras, M.L.; Nussbaum, L.A.; Gurgus, D.; Strat, L.; Zamfir, A.S.; Poroch, V.; Folescu, R. Nosocomial Infections Distribution and Impact in Medical Units. Rev. Chim. 2019, 70, 2265-2268. doi:10.37358/RC.19.6.7319. 

Costinar, L.; Herman, V.; Iancu, I.; Pascu, C. Phenotypic Characterizations and Antimicrobials Resistance of Salmonella Strains Isolated from Pigs from Fattening Farms. Rev. Rom. Med. Vet. 2021, 31(2), 31–34.

In order to adhere to formatting standards and ensure clarity in the presentation of etiological agents, we kindly recommend ensuring that all etiological agents are italicized throughout the entire text, in accordance with international citation conventions in microbiology.

Author Response

Comment 1: line 287: Please clarify limitation, while the study mentions the retrospective design and the absence of MIC values for some years, it could also discuss any potential biases or gaps in the data, especially regarding the generalizability of findings to other regions in Canada or beyond.

Response 1: Thank you for your suggestion regarding our limitations. In new Line 292-308, we had also provided a detailed paragraph regarding the existing limitations of our study aside from retrospective design and missing MIC data.

Although our study did not encompass isolates outside of Fraser Health Authority, we recognize that Fraser Health Authority is the largest health authority in British Columbia, Canada, and encompasses over 75% of annual enteric fever cases in the entire province. In addition, we also mentioned in the Discussion section that our health authority had 118 typhoidal Salmonella isolates (about 1/3 of the isolates reported by the Canadian surveillance program), and therefore represented a significant proportion of Canadian isolates.

We did acknowledge our focus was on blood culture isolates and hence there could be gaps in terms of capturing the absolute full extent for those with less significant infection (e.g., stool culture positive but blood cultures negative). However, bacteremia is typically the most encountered manifestation of enteric fever; blood cultures are considered as a reference standard for diagnosis of enteric fever (Line 304-306). We also knowledge there may be cases that were not captured given lack of confirmed microbiologic diagnosis on blood culture or positive blood cultures outside of Fraser Health (Lines 306-308).

Comment 2: I recommend further Research on XDR S. Typhi, given that the study didn’t observe XDR cases, it would be useful to explore in the discussion section the potential reasons for the absence of such cases in British Columbia compared to Ontario and other regions.

Response 2: Thank you for your response. Thankfully, XDR S. Typhi remains very rare in Canada and we have further described these in our paper with cited sources. There are quite sporadic XDR cases in south Asia of which the largest outbreak was described in Pakistan during November 2016. The transmission of resistance genes via plasmids is certainly complicated and not very well understood why resistant strains may or may not necessarily appear in our Canadian epidemiology. As only 3.5% of typhoidal Salmonella study patients had recently traveled to Pakistan and returned to British Columbia, we wondered if the risk of XDR acquisition/detection was therefore relatively low. Reassuringly, XDR was not detected in our study in a large health authority in British Columbia. In Lines 287-290, we also hypothesize the potential reason why no XDR strains were detected in British Columbia including the potential lack of exposure to circulating outbreak areas in Pakistan.

Furthermore in Lines 284-286, we also provided an update from our CIPARS national (Canadian) surveillance database, which has confirmed there were no further XDR typhoidal Salmonella outside of one individual case in 2023 and none since.

However, given ongoing travel and immigration patterns, there certainly remains an ongoing chance that XDR will eventually be detected in British Columbia, Canada. Thank you again kindly for your feedback.

Comment 3: Please expand on Pre-travel Health Assessments. The suggestions for pre-travel health assessments is important; however, providing more specific recommendations or data on how effective such programs have been elsewhere could strengthen this recommendation.

Response 3: Thank you for your response. We carefully considered how travel medicine is a relatively newer specialty. We read the IDSA travel medicine guidelines which stated that “an awareness has developed among practitioners that prevention of illness in travelers includes not only the provision of vaccines and chemoprophylactics but also a discussion of topics such as personal behavior and safety during travel, prevention of altitude illness, and access to medical care in the event of illness” and that “the application of evidence-based standards to travel medicine is a challenge…The specialty is new and has not had the time required to develop a vast evidence base” (https://doi.org/10.1086/508782).

We performed a literature search using terms such as “pre-travel” and “typhoid” and PubMed. Unfortunately, there was limited data on pre-travel assessments specifically on typhoid; however, this may have been due to an understudied research question. In response, we had included data about typhoid vaccine efficacy in our discussion and included in a reference from the CDC Gold Book on pre-travel assessments (see new cited source 19, Line 245). Given the human-human transmission risk with typhoidal Salmonella, it is certainly recommended to seek pre-travel assessments. Certainly, this opens a future opportunity for new studies to analyze the impact of pre-travel assessments on preventing travel-related infections.

Comment 4: I would like to recommend that you cite the following paper in your study, as it addresses crucial aspects of nosocomial infections and their impact, which aligns with the theme of your research:
This paper provides valuable information on the distribution and frequency of nosocomial infections in healthcare settings. Including it in your references would enhance the comprehensiveness of your literature review and strengthen the foundation for the objectives and conclusions of your study.  

Stanga, L.C.; Vaduba, D.M.B.; Grigoras, M.L.; Nussbaum, L.A.; Gurgus, D.; Strat, L.; Zamfir, A.S.; Poroch, V.; Folescu, R. Nosocomial Infections Distribution and Impact in Medical Units. Rev. Chim. 2019, 70, 2265-2268. doi:10.37358/RC.19.6.7319. 

Costinar, L.; Herman, V.; Iancu, I.; Pascu, C. Phenotypic Characterizations and Antimicrobials Resistance of Salmonella Strains Isolated from Pigs from Fattening Farms. Rev. Rom. Med. Vet. 202131(2), 31–34.

Response 4: Thank you kindly for suggesting the following citations. After careful review by our Team, we have decided not to include the first reference by Stanga et al. Clinically, enteric fever is transmitted by fecal-oral transmission; however, it is not known to be a nosocomial-acquired pathogen at least in developed nations globally. All of our cases captured in this study were presenting from community, with 96% of them having recent travel to an endemic area within. This is in keeping with the enteric fever cases seen in developed nations (including Canada and North America), where there is no known endemic transmission of typhoidal Salmonella.

We also carefully reviewed the second source by Costinar et al. and considered how the pathogen of focus in our paper, typhoidal Salmonella, originates from human reservoirs and is not known to have animal reservoirs (unlike non-typhoidal Salmonella). As such, we considered that this paper was not applicable for our study’s focus. We sincerely thank you for your references.

Comment 5: In order to adhere to formatting standards and ensure clarity in the presentation of etiological agents, we kindly recommend ensuring that all etiological agents are italicized throughout the entire text, in accordance with international citation conventions in microbiology

Response 5: Thank you for going over this in detail. We find Salmonella nomenclature to be quite intricate and interesting. We have consulted CDC system nomenclature, which is based on recommendations from the WHO Collaborating Centre.

According to this nomenclature system (link: https://wwwnc.cdc.gov/eid/page/scientific-nomenclature), serotypes or serovars  (such as Typhi and Paratyphi) are capitalized, but not italicized since they are not technically species names. For example, correct nomenclature will be either Salmonella ser. Typhi or Salmonella enterica subsp. enterica ser. Typhi, or Salmonella Typhi as we indicated in our manuscript. Of note, we did italicize the genus Salmonella, either fully or with the shorthand letter “S.” followed by the serovar.